Efficacy of trimetazidine for myocardial ischemia-reperfusion injury in rat models: a systematic review and meta-analysis

Zhang Xiaobin 1 2
Duan Zhanhui 1
Yu Yanpu 1 2
Li Chunjing 1
Hao Mingyao 1 2
Ma Yuning 1
Ma Yuxia phdmayuxia@yeah.net 1
Du Dongqing ddqsdzyydx11@yeah.net 1
1 Shandong University of Traditional Chinese Medicine , Jinan , Shandong , China
2 Department of Traditional Chinese Medicine External Treatment Center, Affiliated Hospital of Shandong University of Traditional Chinese Medicine , Jinan , Shandong , China
Upadhyay Rohit
Electronic publication date: 2025 Jun 6
Publication date: 2025
Volume: 13
Electronic Location ID: e19515
Received 2024 Dec 16; Accepted 2025 May 2
Copyright: ©2025 Zhang et al.
Copyright year: 2025
Copyright holder: Zhang et al.
License: This is an open access article distributed under the terms of the Creative Commons Attribution License, which permits unrestricted use, distribution, reproduction and adaptation in any medium and for any purpose provided that it is properly attributed. For attribution, the original author(s), title, publication source (PeerJ) and either DOI or URL of the article must be cited.
License URL: https://creativecommons.org/licenses/by/4.0/

Keywords: Myocardial ischemia-reperfusion injury, Trimetazidine, Superoxide dismutase, Creatine kinase isoenzyme, Myocardial infarct size, Systematic review, Meta-analysis, Malondialdehyde, Lactate dehydrogenase

Funding: Shandong University of Traditional Chinese Medicine YJSTZCX2024014 Shandong Province Natural Science Foundation Joint Fund Project ZR2021LZY044 Natural Science Foundation of Shandong Province ZR2021MH373 Jinan “GaoXiao 20 Tiao” Funding Project Contract 2020GXRC005 Qilu Health Leading Talent Project Lu Wei Talent Word [2020] no. 3 NATCM’s Project of High-level Construction of Key TCM Disciplines zyyzdxk-2023116 High-Level Traditional Chinese Medicine Key Disciplines of the State Administration of Traditional Chinese Medicine and External Treatment of Traditional Chinese Medicine The Natural Science Foundation of Shandong Province ZR2023QH102 This study was supported by the 2024 Quality Improvement and Innovation Project for Doctoral Students at Shandong University of Traditional Chinese Medicine (grant no. YJSTZCX2024014), the Shandong Province Natural Science Foundation Joint Fund Project (grant no. ZR2021LZY044), the Natural Science Foundation of Shandong Province (grant no. ZR2021MH373), Jinan “GaoXiao 20 Tiao” Funding Project Contract (grant no. 2020GXRC005), Qilu Health Leading Talent Project, Lu Wei Talent Word [2020] no. 3, NATCM’s Project of High-level Construction of Key TCM Disciplines (grant no. zyyzdxk-2023116), High-Level Traditional Chinese Medicine Key Disciplines of the State Administration of Traditional Chinese Medicine, and External Treatment of Traditional Chinese Medicine, the Natural Science Foundation of Shandong Province (grant no. ZR2023QH102). The funders had no role in study design, data collection and analysis, decision to publish, or preparation of the manuscript.

==============================
Context

Trimetazidine (TMZ) is used as a medication for ischemic heart disease treatment. Recently, several animal models have been studied in relation to the research on myocardial ischemia-reperfusion injury (MIRI) treatment. In this study, we conducted a meta-analysis of TMZ in rat MIRI models to evaluate TMZ’s therapeutic efficacy.

Methods

We systematically searched eight databases for studies on TMZ in rat MIRI models. We utilized two literature quality assessment criteria to evaluate the paper quality. Assessment of TMZ treatment efficacy was based on the outcomes, as well as on the subgroup analysis. This study was registered at PROSPERO (registration number CRD42022377728).

Results

After applying the inclusion and exclusion criteria, 24 eligible studies were shortlisted from 405 studies. We found that, in rat MIRI models, TMZ dramatically boosted the superoxide dismutase (SOD) levels while decreasing the levels of malondialdehyde (MDA), lactate dehydrogenase (LDH), and creatine kinase isoenzyme (CK-MB), and the infarct size. In addition, the duration of myocardial ischemia, reperfusion duration, dosage, rat species and mode of administration influenced the effectiveness of TMZ. The result indicated that TMZ had a considerable therapeutic effect on the duration of myocardial ischemia at less than 30 min as well as on the duration of reperfusion at 120–180 min. In fact, it was more effective when administered intravenously and via gavage at doses of 3–10 mg/kg.

Conclusion

TMZ can attenuate the damage caused by MIRI in rat, with a myocardial protective effect. These findings would facilitate preclinical evidence for further investigation.

Introduction

Cardiovascular disease is associated with the most significant morbidity and death rates worldwide. In 2017, cardiovascular diseases, including acute myocardial infarction, accounted for nearly 178 million deaths across the world (Saglietto et al., 2021). It is believed that the incidence of acute myocardial infarction would continue rising annually with the aging world population. Acute myocardial infarction significantly affects the economic and health burdens of the global population. The most effective strategy to prevent myocardial cell death resulting from myocardial ischemia injury is to promptly restore the coronary blood supply (Zhang et al., 2022a; Zhang et al., 2022b). At this stage of the disease, the coronary blood flow may be restored mainly through revascularization techniques such as thrombolysis and coronary intervention (Ozkalayci et al., 2022). However, ischemia-reperfusion injury can occur in such cases after revascularization due to prolonged ischemic circumstances, which can worsen the extent of myocardial damage.

Myocardial ischemia-reperfusion injury (MIRI) is an inevitable secondary damage occurring in ischemic heart diseases. It is characterized as an increase in the levels of oxygen-free radicals, calcium overload, disruption of the energy metabolism, and apoptosis following the sudden restoration of blood flow to the ischemic myocardium, which can ultimately lead to myocardial contractile dysfunction, structural damage to myocardial cells and cell death, which can prolong myocardial infarction and deteriorate the cardiac functions (Liu et al., 2018; He et al., 2022). MIRI is typically caused by vascular occlusion leading to hypoxia in distal tissues. Hypoxia results in abnormal cellular physiology and ultimately cell death. Prior research has demonstrated a myriad of cellular physiological mechanism which contribute to MIRI including ion accumulation, mitochondrial membrane potential disturbance, reactive oxygen species (ROS) production, dysregulation of nitric oxide metabolism, platelet aggregation, immunological activation, and apoptosis (Heusch, 2020). The precise mechanism of MIRI remains unknown. Modern medicine offers a range of therapies for MIRI, such as antioxidant agents and calcium channel blockers (Rout et al., 2020). However, these therapies are not fully validated or developed, and the quest for safe and effective interventions continues. Therefore, it is essential to identify safe and effective therapeutic options for MIRI.

Trimetazidine (TMZ) was developed in 1964 by Servier Laboratories in France and was initially launched as an anti-anginal drug. TMZ is a 3-ketoacyl-CoA thiolase inhibitor that inhibits cardiomyocyte mitochondrial fatty acid absorption and increases the aerobic glucose metabolism (Knuuttila et al., 2019). By modifying the aerobic metabolic route of cardiomyocytes, boosting the efficiency of ATP synthesis (Li et al., 2017), and maintaining the structure and function of mitochondria, it can exert a direct protective effect at the cellular level. TMZ restores the function of ischemic hearts by inhibiting the ROS/NFκB pathway and modulating the expression of the mitochondrial calcium uniporter (Xiao et al., 2023). TMZ attenuates MIRI-mediated ferroptosis by activating the Sirt3-Nrf2/GPX4/SLC7A11 signaling pathway (Tan et al., 2024). Clinical studies have shown that TMZ is used in the treatment of cardiovascular diseases such as angina pectoris, myocardial infarction, MIRI, chronic congestive heart failure, and diabetic cardiomyopathy, among others (Danchin et al., 2011; Cavar et al., 2016; Zhang et al., 2016). The approval status of TMZ varies globally, with differences in indications and safety restrictions. In several European countries, such as France, Italy, and Spain, it is approved for angina pectoris and vertigo. However, the European Medicines Agency limited its use in 2012 to second-line angina treatment due to potential neurological side effects, such as Parkinsonian symptoms. The UK does not include it in routine treatments. In China, it is widely approved for cardiovascular diseases, including angina, myocardial ischemia, and chronic heart failure. Similarly, in India, it is approved for angina and myocardial ischemia and is widely used. In contrast, TMZ is not FDA-approved and unavailable in the US. These variations result from differing evaluations of its clinical data and safety by regulatory authorities. Rat models with MIRI are most commonly developed by inducing myocardial ischemia through the ligation of the left anterior descending coronary artery with a slipknot and reperfusion of the myocardium by releasing the ligature after some time (Xu et al., 2021). Elevation of the ST-segment on the electrocardiogram indicates successful modeling (Fan et al., 2021).

Oxidative stress is a significant element of MIRI. Superoxide dismutase (SOD) and malondialdehyde (MDA) are important oxidative stress markers. Lactate dehydrogenase (LDH) is broadly distributed in the cardiac tissues and frequently used as a marker of myocardial damage and progression of a myocardial illness (Jiang et al., 2024). The creatine kinase isoenzyme (CK-MB) levels in the myocardium indicates the severity of myocardial ischemia (Li et al., 2020). Myocardial infarct size is the most intuitive assessment in myocardial ischemia studies (Liu et al., 2022).

In this study, we systematically evaluated and meta-analyzed the therapeutic effects of TMZ in rat MIRI models based on extensive experimental animal data so as to provide insightful data support for the application of TMZ in the treatment of MIRI.

Methods

Protocol and registration

The present meta-analysis was conducted in accordance with the Reporting Items for Systematic Reviews and Meta-Analyses (PRISMA) (Wei et al., 2024). The review protocol has been registered with the International Prospective Register of Systematic Reviews (PROSPERO) (registration no. CRD42022377728).

Search strategy

The databases PubMed, Web of Science, Embase, Cochrane Library, China National Knowledge Infrastructure database (CNKI), Wan-fang database (Wanfang), Chinese Scientific Journals database (VIP), and Chinese Biomedicine database (CBM) were searched using computers since inception until June 30, 2024. The papers were restricted to those published in Chinese and English languages. All searches were conducted using subject words and free words.

Two investigators (Xiaobin Zhang and Zhanhui Duan) searched the databases using the following terms, separately, for all relevant articles. The retrieved words included the following “Myocardial reperfusion Injury”, “Injuries, Myocardial Reperfusion”, “Myocardial Reperfusion Injuries”, “Reperfusion Injuries, Myocardial”, “Reperfusion Injury, Myocardial”, “Injury, Myocardial Reperfusion”, “Myocardial Ischemic Reperfusion Injury”, “Myocardial ischemia-reperfusion injury”, “Myocardial ischemia reperfusion injury”, “Rats”, “Rat”, “Rattus”, “Rattus norvegicus”, “Rats, Norway”, “Rats, Laboratory”, “Laboratory Rat”, “Laboratory Rats”, “Rat, Laboratory”, “Trimetazidine”, “TMZ”, “Centrophène”, “Trimetazidine Dihydrochloride”, “Dihydrochloride, Trimetazidine”, “Vastarel”, “Trimétazidine Irex”, “Vasartel”, and “Idaptan”. The search strategy is detailed in Table S1.

Inclusion and exclusion criteria

The inclusion criteria were as follows: (a) rats as the animal species; (b) MIRI model in rats, specifically modeled via ligation of the left anterior descending coronary artery and reperfusion; (c) TMZ as the intervention approach; (d) the intervention with TMZ was compared with a control group administered normal saline, purified water, or blank intervention. There was no restriction on the dose and method of termiticide administration, treatment duration, and follow-up period; (e) randomized controlled trials; (f) outcome measures included any of the following: levels of SOD, MDA, LDH, and CK-MB, and the myocardial infarct size.

The exclusion criteria were as follows: (a) research involved laboratory animals other than rats; (b) the experimental group or control group involved the use of other drugs; (c) studies with inaccurate data extraction or missing data; (d) review articles, clinical case reports, letters, comments, and case reports; (e) does not include test indicators to be evaluated.

Literature selection and data extraction

Two researchers (Xiaobin Zhang and Zhanhui Duan) individually conducted the literature review, data extraction, and cross-checking. In case of debate, a third party (Dongqing Du) was sought for discussion and resolution. The collected data from the enrolled literature included the first author’s name, published time, species, sex, weight, ischemia duration, reperfusion duration, groups, dosage, route, and treatment time. The Endnote X9 software was used to eliminate duplicate literature. The duplicate screening process was conducted by two independent reviewers to ensure quality control. In cases where conflicting decisions arose during the review, a third reviewer was consulted to resolve discrepancies. The process was divided into two stages. During the initial stage, the reviewers performed title and abstract screening. References from the selected articles were manually searched to identify relevant studies, which were then subjected to the same screening procedure. In the subsequent stage, full-text articles were thoroughly reviewed to confirm their compliance with the inclusion criteria.

Risk of bias in individual studies

The SYRCLE’s Risk of Bias tool was employed to evaluate the risk bias of experimental animal research (Hooijmans et al., 2014; Sun et al., 2021). The tool was used to analyze the sequence generation, baseline characteristics, allocation concealment, random housing, blinding, random outcome assessments, incomplete outcome data, selective outcome reporting, and other sources of bias. A response of “yes” indicated that the risk of bias in the study was low, while a response of “no” indicated that the risk of bias was high, and an “unclear” response implied inadequate details to determine the risk of bias in the study.

To assess the quality of the included studies, we used a revised version of the Collaborative Approach to Meta-Analysis and Review of Animal Data from Experimental Studies (CAMARADES) (Sun et al., 2021) checklist. The following were the primary evaluation indicators: peer-reviewed publications, presence of randomization of subjects into treatment groups, assessment of the dose–response relationship, blinded assessment of behavioral outcomes, monitoring of physiological parameters such as body temperature, calculation of the necessary sample size to achieve sufficient power, statement of compliance with the animal welfare regulations, avoidance of anesthetic agents with marked intrinsic neuroprotective properties, statement of potential conflict of interests, and the use of a suitable animal model. Records containing pertinent materials in the included literature were tallied, and the final point total was calculated. Two writers (Xiaobin Zhang and Zhanhui Duan) independently evaluated the quality of the enrolled studies by using the SYRCLE’s risk of bias tool and CAMARADES. In case of any discrepancies, resolution was obtained through mutual discussion.

Statistical analysis

The meta-analysis was conducted using the Review Manager Software (RevMan 5.4) and Stata 16.0 software. Data compilation utilized relative risk (RR) with a 95% confidence interval (CI). Standard mean difference (SMD) or mean difference (MD) and 95% CI were applied to the measurement of data. When the heterogeneity was minimal (p > 0.1, I2 <  50%), a fixed-effect model was implemented to combine statistics; when the heterogeneity was substantial (p ≤ 0.1, I2 ≥ 50%), a random-effect model was employed. Meta-analysis applying the random-effects model was adopted to aggregate RR. Subgroup and meta-regression were conducted to explore the causes of heterogeneity. Sensitivity analysis evaluates the robustness and reliability of the results. Egger’s test and funnel plots were performed to assess any publication bias.

Results

Study selection

As per the article search strategy, 405 articles were retrieved. A total of 189 duplicates were removed using the Endnote X9 software. Then, 125 articles were excluded after reviewing their titles and abstracts. On reading the full article, 67 articles were eliminated. Finally, 24 articles were shortlisted for the present meta-analysis (Fig. 1).

Figure 1 Flow diagram of literature screening.

Study characteristics

This meta-analysis consisted of 24 studies, comprising 16 written in Chinese and eight in English (Kara et al., 2006; Pantos et al., 2005; Kutala et al., 2006; Cheng, Jia & Xu, 2007; Li et al., 2009; Khan et al., 2010; Cai & Gao, 2011; Li, Liu & Guo, 2013; Ruan et al., 2013; Zhou et al., 2013; Li et al., 2014; Şentürk et al., 2014; Zhou et al., 2014; Ma et al., 2016; Zhao & Zhang, 2017; Cheng et al., 2018; Fan et al., 2018; Wu et al., 2018; Zhu, 2018; Zhong et al., 2018; Yu et al., 2019; Qiao, Wang & Sun, 2020; Yu et al., 2020; He et al., 2023). A total of 641 rats were counted, with 313 rats in the experimental group and 328 in the control group. The experimental group received trimetazidine, while the control group was administered 0.9% normal saline, 0.85% normal saline, blank, or purified water. The provided articles outline the use of the left anterior descending coronary artery ligation and subsequent release as methods for establishing a rat model of MIRI. SOD was included in four reported outcome measures, MDA in 10, LDH in 13, CK-MB in 7, and myocardial infarct size in 12. The characteristics of the included articles are detailed in Table S2 .

Quality assessment

With the SYRCLE’s Risk of Bias research and analysis tool, two studies (Pantos et al., 2005; Li et al., 2009) illustrated the randomization method, while other studies only stated the application of randomization without describing the randomization method. Two studies (Li et al., 2009; Ruan et al., 2013) described the random outcome assessment, but others did not. The baseline characteristics of the TMZ and control groups were identified to be the same. All studies presented their findings precisely and thoroughly. None of the studies revealed any other bias and hence deemed “low risk.” None of the studies provided information regarding allocation concealment, whether animals were housed randomly during the experiment, or whether caretakers and/or researchers were blinded; hence they were all rated “unclear” (Fig. 2). Published literature states additional limits in the design and implementation of animal experimental methodology that warrants exploration to ensure the practicability of translating basic research into clinical investigations.

Figure 2 (A–B) Evaluation of literature quality results obtained through SYRCLE’s risk of bias based on the Cochrane tool.

The included studies scored an average of 5.54 out of 10 points on the modified CAMARADES for quality evaluation, demonstrating that the quality of animal experiments need to be improved in terms of uniformity and direction. All enrolled studies were peer-reviewed publications, randomization of subjects into treatment groups, use of a suitable animal model and avoidance of anesthetic agents with marked affect cardiac function. Eleven studies (Kara et al., 2006; Pantos et al., 2005; Li et al., 2009; Khan et al., 2010; Li et al., 2014; Şentürk et al., 2014; Ma et al., 2016; Zhao & Zhang, 2017; Wu et al., 2018; Qiao, Wang & Sun, 2020; He et al., 2023) demonstrated compliance with animal welfare regulations. No studies computed the sample size necessary to meet the stated test quality. All studies employed appropriate animal models. Fifteen studies (Kara et al., 2006; Pantos et al., 2005; Cheng, Jia & Xu, 2007; Khan et al., 2010; Cai & Gao, 2011; Li, Liu & Guo, 2013; Şentürk et al., 2014; Zhou et al., 2014; Ma et al., 2016; Zhao & Zhang, 2017; Fan et al., 2018; Wu et al., 2018; Zhu, 2018; Zhong et al., 2018; He et al., 2023) were conducted to evaluate physiological parameters. Six studies (Cheng, Jia & Xu, 2007; Zhou et al., 2013; Şentürk et al., 2014; Zhou et al., 2014; Fan et al., 2018; Zhu, 2018) illustrated the assessment of the dose–response relationship. One study (Ruan et al., 2013) described blinded assessment of behavioral outcomes. Four studies (Cai & Gao, 2011; Li et al., 2014; Wu et al., 2018; Şentürk et al., 2014) declared no potential conflicts of interests (Table S3).

Meta-analysis of primary outcomes

Superoxide dismutase

Four of the included studies (Li et al., 2009; Zhou et al., 2014; Zhong et al., 2018; Yu et al., 2019) reported the impact of TMZ on the SOD levels’ intervention in rat models with MIRI (experimental group, n = 62; control group, n = 62). Considering the heterogeneity of the assessed articles (p = 0.002, I2 = 76%), we utilized a random-effects model to pool the data from six studies. The TMZ group showed significantly increased SOD levels relative to that in the control group, according to this meta-analysis [MD = 36.63, 95% CI [24.66–48.61], p < 0.001] (Fig. 3A). Because of inter-study heterogeneity, we also performed sensitivity analysis by removing each study one by one, and the results were robust (Fig. 3B). To further clarify the sources of heterogeneity, we performed subgroup analysis and meta-regression. Subgroup analysis and meta-regression based on gender distribution (male or male and female), ischemia duration (time < 40 min or 40 min ≤ time ≤ 90 min), reperfusion duration (30 min ≤ time < 120 min, 120 min ≤ time < 180 min or 180 min ≤ time ≤ 480 min), dosage (10 mg kg−1 d−1 ≤ dosage < 20 mg kg−1 d−1 or 20 mg kg−1 d−1 ≤ dosage ≤ 540 mg kg−1 d−1), treatment time (before ischemia or before reperfusion), and rat species (SD or Wistar). Meta-regression analyses did not identify a source of heterogeneity in SOD (Table S4). The results of subgroup analysis demonstrated that heterogeneity was significantly reduced in the subgroups of male and female gender distribution, ischemic duration less than 40 min, reperfusion duration of 120–180 min, treatment time before reperfusion, rat species (Table S5 ) (Figs. S1–S6), suggesting that these elements may be a source of heterogeneity. Importantly, however, considerable heterogeneity was noted even in the remaining subgroups (Table S5) (Figs. S1–S6), implying that heterogeneity may have been derived from other sources. As SOD was affected by numerous variables, we considered that variations in the testing equipment and methods adopted across the different researches may be one of the primary causes.

Figure 3 (A) Forest plot of SOD; (B) sensitivity analysis of SOD.

Malondialdehyde

Ten studies (Kara et al., 2006; Cheng, Jia & Xu, 2007; Li et al., 2009; Cai & Gao, 2011; Şentürk et al., 2014; Zhou et al., 2014; Cheng et al., 2018; Wu et al., 2018; Zhong et al., 2018; Yu et al., 2019) explored the effect of TMZ on the MDA levels following intervention in rat models with MIRI (experimental group, n = 142; control group, n = 156). Considering the heterogeneity of the results (p < 0.001, I2 = 100%), we utilized a random-effects model. The present meta-analysis revealed that the TMZ group could considerably lower the level of MDA relative to that in the control group [MD = −0.17, 95% CI [−0.21 to −0.12], p < 0.001] (Fig. 4A). Due to inter-study heterogeneity, we performed sensitivity analysis by removing each study one by one that did not affect the overall estimate of effect (Fig. 4B). We executed meta-regression and subgroup analyses to determine the origins of heterogeneity according to gender distribution (male or male and female), ischemia duration (time < 40 min or 40 min ≤ time ≤ 90 min), reperfusion duration (30 min ≤ time < 120 min, 120 min ≤ time < 180 min, or 180 min ≤ time ≤ 480 min), dosage (three mg kg−1 d−1 ≤ dosage < 10 mg kg−1 d−1, 10 mg kg−1 d−1 ≤ dosage < 20 mg kg−1 d−1 or 20 mg kg−1 d−1 ≤ dosage ≤ 540 mg kg−1 d−1), routes (intravenous [i.v], intragastric [i.g], intraperitoneal [i.p]),treatment time (before ischemia, before reperfusion or during ischemia), and rat species (SD or Wistar). The results of meta-regression analyses did not reveal the source of heterogeneity among studies (Table S4). The results of subgroup analysis revealed that heterogeneity was reduced considerably in the subgroups of male and female gender distribution, during ischemia (Table S6) (Figs. S7–S13), indicating that these factors may be the cause of heterogeneity. Nevertheless, substantial heterogeneity remained in the other subgroups (Table S6) (Figs. S7–S13), implying that heterogeneity may have stemmed from other sources.

Figure 4 (A) Forest plot of MDA; (B) sensitivity analysis of MDA.

Meta-analysis of secondary outcomes

Lactate dehydrogenase

Thirteen of the included studies (Pantos et al., 2005; Kutala et al., 2006; Ruan et al., 2013; Li et al., 2014; Ma et al., 2016; Zhao & Zhang, 2017; Cheng et al., 2018; Fan et al., 2018; Wu et al., 2018; Zhu, 2018; Qiao, Wang & Sun, 2020; Yu et al., 2020; He et al., 2023) reported the impact of TMZ on LDH intervention in rat models with MIRI (experimental group, n = 177; control group, n = 180). Considering that the result was heterogeneous (p <  0.001, I2 = 100%). We accordingly employed a random-effects model, which revealed that, compared to the control group, the TMZ group could significantly lower the LDH levels [MD = −19.16, 95% CI [−20.51 to −17.80], p < 0.001] (Fig. 5A). On account of inter-study heterogeneity, we performed a sensitivity analysis to exclude any study that did not affect the overall estimate of effect (Fig. 5B). Meanwhile, we conducted meta-regression and subgroup analysis to determine the origins of heterogeneity based on gender distribution (male, female or male and female), ischemia duration (time < 40 min or 40 min ≤ time ≤ 90 min), reperfusion duration (30 min ≤ Time < 120 min, 120 min ≤ time < 180 min or 180 min ≤ time ≤ 480 min), dosage (10 mg kg−1 d−1 ≤ dosage < 20 mg kg−1 d−1, 20 mg kg−1 d−1 ≤ dosage ≤ 540 mg kg−1 d−1, 10 µmol L−1 ≤ Dosage ≤ 50 µmol L−1), route (i.g, i.p, or perfusion), treatment time (before ischemia, before reperfusion, after reperfusion, during ischemia and reperfusion, during reperfusion, or prior to ischemia and prior to reperfusion), experiment type (in vivo or ex vivo), and rat species (SD or Wistar). The results of meta-regression analyses revealed that reperfusion duration and route of administration may be a source of high heterogeneity (Table S4). Subgroup analyses also confirmed better homogeneity with after reperfusion (Table S7) (Figs. S14–S21), implying that this factor may be the source of heterogeneity. However, substantial heterogeneity remained in the other subgroups (Table S7) (Figs. S14–S21), implying that heterogeneity may have stemmed from additional sources.

Figure 5 (A) Forest plot of LDH; (B) sensitivity analysis of LDH.

Creatine kinase isoenzyme

Seven of the included studies (Kara et al., 2006; Li, Liu & Guo, 2013; Zhao & Zhang, 2017; Cheng et al., 2018; Wu et al., 2018; Zhong et al., 2018; Yu et al., 2020) reported the impact of TMZ on CK-MB intervention in rat models with MIRI (experimental group, n = 82; control group, n = 96). Considering the heterogeneity of the results (p < 0.001, I2 = 99%), we applied a random-effects model in this case. When compared to the control group, the TMZ group significantly lowered the CK-MB levels [MD = −0.77, 95% CI [−1.27 to −0.27], p = 0.003] (Fig. 6A). Because of inter-study heterogeneity, we performed a sensitivity analysis. When each included study was excluded one by one, the effect size did not change significantly, indicating the stability of the results (Fig. 6B). Besides, we executed meta-regression and subgroup analyses to determine the origin of heterogeneity in relation to gender distribution (male or female), ischemia duration (time < 40 min or 40 min ≤ time ≤ 90 min), reperfusion duration (120 min ≤ time < 180 min or 180 min ≤ time ≤ 480 min), dosage (10 mg kg−1 d−1 ≤ dosage < 20 mg kg−1 d−1 or 20 mg kg−1 d−1 ≤ dosage ≤ 540 mg kg−1 d−1), route (i.v, i.g or i.p), treatment time (before ischemia or before reperfusion), and rat species (SD or Wistar). Meta-regression analyses did not reveal the source of heterogeneity in CK-MB (Table S4). Subgroup analyses demonstrated that heterogeneity decreased significantly in the subgroup with an ischemia duration of 40–90 min (Table S8) (Figs. S22–S28), suggesting that this factor may be the origin of heterogeneity. Nonetheless, considerable heterogeneity was noted in the remaining subgroups (Table S8) (Figs. S22–S28), indicating that heterogeneity may have stemmed from other sources.

Figure 6 (A) Forest plot of CK-MB; (B) sensitivity analysis of CK-MB.

Myocardial infarct size

Twelve of the included studies (Kara et al., 2006; Kutala et al., 2006; Khan et al., 2010; Ruan et al., 2013; Zhou et al., 2013; Li et al., 2014; Şentürk et al., 2014; Ma et al., 2016; Wu et al., 2018; Zhong et al., 2018; Qiao, Wang & Sun, 2020; He et al., 2023) reported the impact of TMZ on the myocardial infarct size intervention in rat models with MIRI (experimental group, n = 155; control group, n = 169). Considering the result being heterogeneous (p < 0.001, I2 = 99%), we applied a random-effects model. Compared to the control group, the TMZ group showed significantly lower infarct size [MD =−12.33, 95% CI [−15.38 to −9.28], p < 0.001] (Fig. 7A). Owing to inter-study heterogeneity, sensitivity analysis was conducted. Sensitivity analysis manifested that there were few differences in the pooled effect size estimates and the outcomes were robust (Fig. 7B). Furthermore, we conducted meta-regression and subgroup analyses to determine the origins of heterogeneity as per the gender distribution (male or male and female), ischemia duration (time < 40 min or 40 min ≤ time ≤ 90 min), reperfusion duration (30 min ≤ Time < 120 min, 120 min ≤ time < 180 min or 180 min ≤ time ≤ 480 min), dosage (three mg kg−1 d−1 ≤ dosage < 10 mg kg−1 d−1, 10 mg kg−1 d−1 ≤ dosage < 20 mg kg−1 d−1 or 20 mg kg−1 d−1 ≤ dosage ≤ 540 mg kg−1 d−1, 10 µmol L−1 ≤ Dosage ≤50 µmol L−1), route (i.v, i.g, i.p, or perfusion), treatment time (before ischemia, before reperfusion, after reperfusion, during ischemia, or prior to ischemia and prior to reperfusion), experiment type (in vivo or ex vivo), and rat species (SD or Wistar). Meta-regression results did not reveal the source of heterogeneity among studies (Table S4). Subgroup analyses demonstrated that heterogeneity decreased in the subgroup with a gender distribution of male and female, after reperfusion, during ischemia, and the dosage of 3–10 mg/kg/d (Table S9) (Figs. S29–S36), implying that this factor may be the origin of heterogeneity. Nonetheless, considerable heterogeneity was noted in the remaining subgroups (Table S9) (Figs. S29–S36), implying that heterogeneity may have stemmed from other sources.

Figure 7 (A) Forest plot of myocardial infarct size; (B) sensitivity analysis of myocardial infarct size.

Publication bias analysis

The meta-analysis of the levels of SOD, MDA, LDH, and CK-MB, and the myocardial infarct size was analyzed for publication bias using funnel plots and Egger’s test, respectively. The results revealed that funnel plots and Egger’s tests for each of the factors SOD, MDA, LDH, and CK-MB, and myocardial infarct size displayed poor symmetry in the distribution of the literature, all indicating a degree of publication bias, which may be related to factors such as the low quality of the included articles (Fig. 8). Therefore, publication bias may lead to an overestimation of effect sizes.

Figure 8 (A-J) Egger’s tests and funnel plots for evaluating publication bias.

Discussion

This systematic review and meta-analysis investigated the success elements of animal studies by systematically reviewing studies involving the use of TMZ for MIRI treatment in rats, summarizing the drug’s efficacy and utility, and offering solid evidence for future MIRI treatments. TMZ enhances ATP production efficiency by inhibiting 3-ketoacyl-CoA thiolase, shifting cardiomyocyte energy metabolism from fatty acid oxidation to glucose oxidation. This metabolic regulation differs from other cardioprotective agents. Ischemic preconditioning mitigates prolonged ischemia damage by activating endogenous protective mechanisms, such as the adenosine signaling pathway, through brief ischemia-reperfusion stimuli (Sorge et al., 2024). In contrast, TMZ directly targets cellular metabolism, providing a more direct and controllable protective strategy. Rat was chosen as the research subjects in this study because they serve as a classic model for cardiovascular diseases. Their cardiac structure and function are relatively similar to those of humans, they are convenient for experimental manipulation, their MIRI model exhibits strong reproducibility, their breeding costs are relatively low, and there is extensive experimental data supporting their use. In this study, the left anterior descending coronary artery of the rat models with MIRI was used as the ligature release site to confirm the validity and reasonableness of the input data, which could reduce bias. SOD is the primary enzyme involved in the oxygen radical scavenging mechanism of the body, and MDA is a measure of the extent of free radical damage. In MIRI, the cellular oxidative stress equilibrium is disturbed, and both SOD and MDA levels can be employed as markers of this equilibrium in the body (Zhang et al., 2022a; Zhang et al., 2022b). MIRI can increase cell membrane permeability, leading to the release of intracellular CK-MB and LDH into the blood and a rapid surge in their activities. Consequently, serum LDH and CK-MB can serve as significant biomarkers of myocardial damage (Wang et al., 2018). Meanwhile, myocardial infarct size can clearly demonstrate the extent of myocardial damage. Therefore, we conducted a systematic review and meta-analysis to gather data on TMZ therapy in MIRI rats.

Here are some research findings regarding TMZ. As reported, TMZ features prominently in cardiovascular therapy by enhancing energy metabolism, as demonstrated in the study by Marzilli et al. (2019). TMZ has also been reported to improve the myocardial metabolism directly by modulating beta oxidation (Kantor et al., 2000). Moreover, TMZ induces an increase in the expression of NADPH oxidase 2, while concurrently diminishing the levels of ROS (Shu et al., 2021). TMZ can impede the synthesis of connective tissue growth factor (CTGF) by reducing the production of ROS. This, in turn, leads to a reduction in the accumulation of collagen I and collagen III in myocardial tissues, ultimately achieving amelioration of myocardial interstitial fibrosis (Zhang et al., 2020). TMZ triggers an increase in the production of miR-21, which inhibits PTEN activity, leading to the activation of the PI3K-Akt signal pathway. Consequently, this hinders the expression of Bax/Bcl-2 and caspase-3, ultimately impeding the apoptosis pathway of cardiomyocytes (Ouyang et al., 2024). As revealed by Wu et al. (2018), after the activation of the Akt/mTOR signaling pathway by TMZ, excessive autophagy in the heart can be suppressed, resulting in a positive impact on the heart. TMZ has the additional capacity to inhibit the occurrence of excessive autophagy by down-regulating LC3 and Beclin-1, while also up-regulating the expression of PI3K, p-Akt, and p-GSK-3β. This modulation leads to the alleviation of MIRI and protection on the damaged myocardium by regulating the PI3K/Akt/GSK-3β signaling pathway (Shi et al., 2020). TMZ can activate Akt, promoting the nuclear translocation of HSF1. Through its interaction with the promoter region of VEGF-A, HSF1 stimulates the expression of VEGF-A, culminating in enhanced myocardial angiogenesis and improved cardiac function (Shu et al., 2022).

Oxidative stress is a major pathogenic factor in myocardial injury. SOD removes free radicals and prevents the production of more potent hydroxyl radicals (Geng et al., 2022). Myocardial ischemia and hypoxia can lead to the generation of a large amount of oxygen radicals, triggering an increase in cardiac cell membrane permeability and even necrosis. Increased SOD levels can increase the antioxidant capacity and thus reduce the extent of myocardial damage in the body. Therefore, SOD was chosen as among the primary endpoints in this study. According to the results, TMZ treatment of the rat models with MIRI increased serum SOD levels. In a study, TMZ enhanced SOD activity by activating the SIRT1-AMPK pathway through increased SIRT1 expression and AMPK phosphorylation (Luo et al., 2021). Similar outcomes were observed in the subgroups of gender distribution, ischemia duration, reperfusion duration, dosage treatment time, and rat species. Additionally, we discovered considerable heterogeneity in the results of this analysis. Although our choice of randomized models for data statistics, sensitivity analyses, meta-regression analyses, and the subgroup analysis helped in minimizing the subgroup heterogeneity, still some subgroups exhibited significant heterogeneity, prompting us to be cautious in interpreting the results. We hypothesized that this may due to measurement errors at various research organizations, variations in TMZ formulations, strain-specific responses in rat, and other factors.

MDA is the last product of lipid peroxidation by oxygen radicals. It can induce cell membrane degeneration and a change in membrane fluidity and permeability, inferring the oxygen radical damage level to cardiac myocytes (Nehra et al., 2022). SOD can be used to indicate the body’s antioxidant capacity, while MDA can be used to evaluate free radical levels. Variations in SOD and MDA levels can reflect the equilibrium between the generation and removal of free radicals in the heart muscle. Hence, MDA was also selected as one of the primary study outcomes. The results demonstrated that TMZ could lower serum MDA levels in the rat models with MIRI, with consistent findings noted across subgroups of ischemia duration and route. TMZ inhibits oxidative stress by inducing the inactivation of Nrf2/HO-1 and NF-κB signaling pathways, which reduces MDA levels (Zhang et al., 2019). Nevertheless, when the subgroup analysis was conducted by gender distribution, reperfusion duration, dosage, and treatment time, TMZ did not substantially reduce MDA levels in the subgroup with gender of male and female, reperfusion duration between 180 and 480 min [MD = −0.94, 95% CI [−2.07–0.19], p = 0.10], dosage ranging from 10 to 20 mg/kg [MD = −0.05, 95% CI [−0.11–0.01], p = 0.11], and treatment time is before reperfusion. The results indicated that TMZ might not be sufficiently efficient in enhancing myocardial ischemia-reperfusion for longer reperfusion lengths, lower dosages, and drugs used before reperfusion. However, fewer studies were included in these four subgroups, and this conclusion must be validated in future studies. Furthermore, meta-regression analyses did not reveal the source of heterogeneity among studies. Subgroup analysis discovered that gender and treatment time might cause heterogeneity and that subgroup analysis only partially decreased heterogeneity. Accordingly, we speculated that this could result from discrepancies in measurements among various research organizations, differences in TMZ preparations, rat strain-specific reactions, and additional contributing elements.

Cardiac enzymes are enzymes located within heart muscle cells. When these muscle cells are damaged, cardiac enzymes are released into the blood, thereby reflecting as an increase in the findings of several diagnostic tests. LDH and CK-MB are typical indicators of cardiac enzyme profiles that possibly represent the extent of myocardial ischemic damage (Zhao et al., 2021). TMZ lowered serum LDH levels in the rat models with MIRI, indicating that this drug can effectively prevent myocardial damage and play a role in myocardial protection. Meanwhile, consistent results were achieved in several subgroups of gender distribution, ischemia duration, reperfusion duration, dosage, route, experiment type, and rat species. Nevertheless, the subgroup analysis revealed that the TMZ group exhibited no substantially reduction in serum LDH levels compared with the control group in the subgroups by after reperfusion [MD = −16.80, 95% CI [−57.44–23.84], p = 0.42]. This subgroup consisted of only two studies; hence, this conclusion must be validated through further research. Moreover, the results of subgroup and meta-regression analyses indicated that reperfusion duration, route of administration and treatment time may lead to heterogeneity. Nonetheless, unexplained variability still persisted, which may be attributed to differences in detection equipment and methods across studies, variations in TMZ formulations, strain-specific responses in rat, and other factors. CK-MB is among the most widely used indicators for determining myocardial damage. During myocardial injury, cardiomyocytes produce a significant number of cardiac enzymes, and CK-MB levels in the peripheral blood are abnormally elevated. According to the meta-analysis results, TMZ decreased serum CK-MB levels in the rat models with MIRI. Because of the significant degree of heterogeneity, subgroup analyses were conducted on the basis of gender distribution, ischemia duration, reperfusion duration, dosage, route, treatment duration, and rat species. When the subgroup analysis was conducted by ischemia duration, the results for the 40–90 min subgroup were negative [MD = −8.92, 95% CI [−32.82–14.99], p = 0.46]. When the subgroup analysis was conducted by reperfusion duration, the results for the 30–120 min and 180–480 min subgroup were negative [MD = −0.05, 95% CI [−0.23–0.13], p = 0.59] [MD = −0.40, 95% CI [−0.82–0.02], p = 0.06]. In the subgroup analysis by dosage, the results were negative for subgroups with dosages ranging from 20 to 540 mg/kg [MD = −0.54, 95% CI [−1.17–0.09], p = 0.09]. Upon subgrouping according to the administration route, the findings for the i.p injection and i.v injection subgroup were negative [MD = −60.31, 95% CI [−178.67–58.05], p = 0.32] [MD = −0.05, 95% CI [−0.23–0.13], p = 0.59]. Upon subgrouping according to the treatment time, the findings for the prior to reperfusion subgroup were negative [MD = −0.79, 95% CI [−1.86–0.27], p = 0.14]. After subgroup analysis based on rat species, the results indicated a negative effect in the Wistar subgroup [MD = −503.63, 95% CI [−1,402.41–395.15], p = 0.27]. The ischemia duration, reperfusion duration, TMZ dose, administration route, rat species and treatment time can considerably impact the TMZ efficacy in treating MIRI. According to the results of the subgroup analysis, ischemia duration may lead to heterogeneity. Meta-regression analyses did not identify the source of heterogeneity. Nonetheless, some unexplained heterogeneity was still observed, which may be attributable to differences in detection equipment and methods among experiments, variations in TMZ formulations, strain-specific responses in rats, and so on.

Myocardial infarct size is a crucial index for evaluating the degree of myocardial damage and the effect of drug treatment. It is the most intuitive evaluation index for myocardial ischemia studies. According to the meta-analysis results, TMZ decreased the infarct size in the rat models with MIRI. In some studies, TMZ induced activation of the extracellular signal-regulated kinase (ERK) signaling pathway in the cardiomyocytes of rat with myocardial infarction. This activation increased the expression of phosphorylated ERK (p-ERK), decreased the content of ROS in cardiomyocytes, reduced the expression of apoptotic proteins, decreased the myocardial infarct size, and improved cardiac function (Liu et al., 2016). Because of the significant degree of heterogeneity, the subgroup analyses were conducted on the basis of gender distribution, ischemia duration, reperfusion duration, dosage, route, treatment duration, and rat species. When the subgroup analysis was conducted by route, the results for the perfusion subgroup were negative [MD = −17.20, 95% CI [−37.27–2.87], p = 0.09]. Upon subgrouping according to the experiment type, the findings for the ex vivo subgroup were negative [MD = −17.20, 95% CI [−37.27–2.87], p = 0.09]. Experiment type is hypothesized to considerably impact the TMZ efficacy in treating MIRI. Meta-regression analysis did not identify the source of heterogeneity. According to the subgroup analysis results, gender distribution, dosage, and treatment time may serve as a source of heterogeneity. Nonetheless, unexplained heterogeneity still persisted, which could be attributed to differences in detection equipment and methods among experiments, variations in TMZ formulations, strain-specific responses in rat, and other factors.

This was the first systematic review of TMZ’s effects on rat models with MIRI. The articles included in this review were subjected to a comprehensive screening and control procedure. We extrapolated that the general quality of the included articles was not particularly satisfactory. Despite rigorous evaluating the impact of TMZ intervention on the rat models with MIRI, this study had certain drawbacks. The study shortcomings are as follows: first, articles in only two languages were searched: Chinese and English. The search for gray literature and conference papers was insufficient because acquiring them was difficult. Second, some outcomes were highly heterogeneous, the number of articles included were small, and the outcome indicators exhibited some publication bias. Third, this article was an animal experiment-based meta-analysis review, which might be influenced by the quality of the original study design, resulting in a poorer quality of the article as a whole. This might affect the accuracy of the results. Fourth, the common limitations of these studies were a lack of describing allocation concealment, blinding for caregivers and/or investigators, and blinding of outcome assessment. Hence, the included studies were classified as unclear-risk. The absence of these methods may compromise the scientific rigor and reliability of the results in experimental research. A bigger sample size and higher-quality studies are required to validate the results and establish the trustworthiness of the study conclusions. Future studies should strictly follow the CAMARADES evaluation and strictly control the literature inclusion criteria to avoid incorporating studies with high risk of bias and low-quality literature.

Conclusion

A total of 24 articles were enrolled in this meta-analysis based on their meeting the inclusion criteria. Analysis of the levels of SOD, MDA, LDH, and CK-MB, and the myocardial infarct size revealed credible preclinical evidence. A thorough examination of the application of TMZ in MIRI rats revealed that the best dose regimen is 3–10 mg/kg, the most efficient administration route is through i.v injection and gavage, the best ischemia time window is less than 40 min, reperfusion time window is 120–180 min, and the best indicative outcomes are increase in the serum SOD level and decrease in the serum levels of MDA, LDH, and CK-MB, as well as reduced infarct size. Therefore, it will be of significant reference value for the application of TMZ in the clinical treatment of MIRI. However, some of the studies included in this research had relatively small experimental sample sizes and were all based on animal models. Additionally, a limitation of this study is that the effects were only validated in rat models, and there are significant differences between rats and humans. Therefore, further clinical trials are needed in the future to verify the accuracy of the research findings.

Supplemental Information

Supplemental Information 1 Supplementary Figures

Supplemental Information 2 Detailed database search strategy

Supplemental Information 3 Characteristics of the included studies

Supplemental Information 4 CAMARADES evaluation results

Supplemental Information 5 Results of meta regression analysis

Supplemental Information 6 Subgroup analysis of SOD based on gender distribution, ischemia duration, reperfusion duration, dosage, and treatment time

Supplemental Information 7 Subgroup analysis of MDA based on gender distribution, ischemia duration, reperfusion duration, dosage, route, and treatment time

Supplemental Information 8 Subgroup analysis of LDH based on gender distribution, ischemia duration, reperfusion duration, dosage, route, treatment time, and experiment type

Supplemental Information 9 Subgroup analysis of CK-MB based on gender distribution, ischemia duration, reperfusion duration, dosage, route, and treatment time

Supplemental Information 10 Subgroup analysis of myocardial infarct size based on gender distribution, ischemia duration, reperfusion duration, dosage, route, treatment time, and experiment type

Supplemental Information 11 PRISMA checklist

Supplemental Information 12 Audience for this article

Supplemental Information 13 Systematic review protocol

Additional Information and Declarations

Competing Interests

Author Contributions

Data Availability

The authors declare there are no competing interests.

Xiaobin Zhang conceived and designed the experiments, performed the experiments, analyzed the data, authored or reviewed drafts of the article, and approved the final draft.

Zhanhui Duan analyzed the data, prepared figures and/or tables, authored or reviewed drafts of the article, and approved the final draft.

Yanpu Yu analyzed the data, prepared figures and/or tables, and approved the final draft.

Chunjing Li analyzed the data, prepared figures and/or tables, and approved the final draft.

Mingyao Hao performed the experiments, analyzed the data, authored or reviewed drafts of the article, and approved the final draft.

Yuning Ma performed the experiments, authored or reviewed drafts of the article, and approved the final draft.

Yuxia Ma performed the experiments, authored or reviewed drafts of the article, and approved the final draft.

Dongqing Du conceived and designed the experiments, authored or reviewed drafts of the article, and approved the final draft.

The following information was supplied regarding data availability:

This is a systematic review/meta-analysis.

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
