# Peer review of "Efficacy of trimetazidine for myocardial ischemia-reperfusion injury in rat models: a systematic review and meta-analysis"

_PeerJ, doi:10.7717/peerj.19515_

## Round 0.1 · original submission · Major Revisions

Please respond to the comments of the reviewers. In particular, Reviewers 1 and 3 have some serious concerns which must be addressed adequately

Reviewer 1 ·

Basic reporting

While the review/meta-analysis has been executed accurately, I have reservations with regard to the interest and overall utility of this article to contribute significantly to new understanding or insight of a therapeutic developed over 4 decades ago; including the current application towards myocardial ischemia-reperfusion (MIR) injury. Please see my comments below.

Comments:
1. While the retrospective analysis of TMZ based on several biomarkers of oxidative stress and myocardial injury was extensive, the meta-analysis of such a niche area of interest and the lack of true therapeutic efficacy observed through the retrospective analysis would limit the readership and does not “…facilitate decision-making in MIRI treatment strategies.”.
2. TMZ has limited use clinically due to potential side effects, most notably the development of Parkisonian symptoms, which can outweigh its benefits in treating angina. Furthermore, advisory bodies including the European Medicines Agency (EMA) has recommended that it only be used as adjunctive therapy (second-line) if angina is not controlled by other more efficacious therapeutics. Moreover, The U.S. Food and Drug Administration has not approved the use of TMZ in the United States, given its neurological side effects and its inferiority to other available medications to treat angina.
3. Written manuscript is a bit ambiguous in some statements and sentence structure. For example, within the Introduction: Lines 67-72 “The etiology of MIRI is complicated…” The cause of MIRI is quite simple, there is normally an occlusion limiting the oxygen to the distal tissue. Lack of oxygen leads to abnormal cell physiology and death. The following statement should read, “Prior research has demonstrated a myriad of cellular physiological mechanism which contribute to MIRI including….” Sentence discussing “Modern medicine offers…” (line 71-72) is confusing; antioxidant stress is not a therapeutic as written by the authors. Please correct all English language and sentences structure.

Experimental design

1. Introduction is insuffiecient, should include further details regarding the history of TMZ and its utility (or lack thereof) for cardiovascular related diseases and the discrepancy between countries approval of the medication.
2. The authors do not discuss or take into account the species of rat used with the 24 studies which were analyzed. As with most rodent studies, there is significant variability between species of rats (i.e. sprague-dawley v. wistar). This should be discussed.

Validity of the findings

1. Surveying methodlogy is sufficient.
2. Sources seem to be adequately cited, paper is a bit redundant when comparing to prior PeerJ publication by the same group. https://peerj.com/articles/17885/ PMID: 39161965.
3. Organization of the review is good.
4. I do not believe the meta-analysis performed provides sufficient evidence to promote further research on TMZ in MIRI as suggested by the authors in the final sentence of the introduction or in the conclusion. Far superior pharmacological agents are available for angina pectoris. SOD mimetics as ell as other modulators of oxidative stress have all been tested with limited or neutral outcomes.

·

Basic reporting

Dear authors,
The manuscript was written clear and good. It focused common issue. I have some advices becoming more qualified given below.
-In introduction part, TMZ effected pathways should be explained more detailly.
- The reason of rat choosing shold be explain

Experimental design

It was very clear and systematic.

Validity of the findings

no comment

Additional comments

no comment

Reviewer 3 ·

Basic reporting

General Comments
This systematic review and meta-analysis evaluates the therapeutic efficacy of trimetazidine (TMZ) in rat models of myocardial ischemia-reperfusion injury (MIRI). The study synthesizes preclinical evidence across 24 studies, highlighting TMZ’s potential to mitigate oxidative stress, reduce infarct size, and improve cardiac biomarkers. While the manuscript is comprehensive, several areas require clarification or improvement to strengthen validity and readability.

Experimental design

Major Strengths
1. Rigorous Methodology:
- Adherence to PRISMA guidelines and PROSPERO registration enhances transparency.
- Inclusion of both English and Chinese databases reduces language bias.
- Use of SYRCLE’s Risk of Bias tool and CAMARADES checklist ensures systematic quality assessment.
2. Robust Statistical Analysis:
- Appropriate use of random-effects models to account for high heterogeneity.
- Sensitivity and subgroup analyses explore sources of variability (e.g., ischemia duration, dosage).
3. Clinical Relevance:
- Identifies optimal TMZ dosing (3–10 mg/kg), administration routes (i.v., i.g.), and treatment windows (ischemia <30 min, reperfusion 120–180 min), providing actionable insights for future research.
Major Weaknesses and Recommendations
1. High Heterogeneity and Publication Bias:
- Issue: Significant heterogeneity (\(I^2 = 76–100\%\)) across outcomes (e.g., MDA, LDH) and publication bias (asymmetric funnel plots) undermine result reliability.
- Recommendation: Discuss potential reasons for heterogeneity beyond methodological variability (e.g., differences in TMZ formulations, strain-specific responses). Acknowledge that publication bias may overestimate effect sizes.
2. Risk of Bias in Included Studies:
- Issue: Most studies lacked details on allocation concealment, blinding, and randomization methods (rated “unclear risk”).
- Recommendation: Highlight how these limitations affect translational validity. Suggest stricter inclusion criteria (e.g., excluding studies with high bias) in future updates.
3. Animal species selection:
- Issue: One of the inclusion criteria is research involving rats.
- Recommendation: Explain the reason for focusing on rats rather than other laboratory animals
Minor Revisions
1. Methods:
- Specify how duplicate screening and data extraction were resolved (e.g., third-party arbitration).

Validity of the findings

Major Weaknesses and Recommendations
1. Terminology and Clarity:
- Issue: “Methane dicarboxylic aldehyde” is non-standard; use “malondialdehyde (MDA)” for consistency with literature.
- Recommendation: Revise terminology and streamline the introduction to avoid redundancy (e.g., repetitive background on oxidative stress).
2. Clinical Translation:
- Issue: Overemphasis on TMZ’s clinical potential without cautioning about preclinical limitations.
- Recommendation: Add a paragraph on the necessity of clinical trials to validate findings, noting species differences and TMZ’s existing clinical use in other contexts.

Additional comments

Minor Revisions
1. Abstract:
- Clarify that findings are preclinical. Replace “decision-making in MIRI treatment strategies” with “preclinical evidence for further investigation.”
2. Results:
- Report exact \(p\)-values for heterogeneity tests (e.g., “\(p < 0.001\)” instead of “\(p < 0.00001\)”).
3. Discussion:
- Compare TMZ’s efficacy with other cardioprotective agents (e.g., ischemic preconditioning) to contextualize its novelty.
4. Abbreviations:
- Check the abbreviations carefully (e.g. LDH is Lactate dehydrogenase)

---

## Round 0.2 · accepted · Accept

Authors have addressed all of the reviewers' comments and manuscript is ready for publication.

Reviewer 3 ·

Basic reporting

Clarity and Professional English: The manuscript is well-written in clear, professional English with minimal grammatical errors. The language is appropriate for an international audience.

Literature References and Background: The introduction provides a comprehensive background on myocardial ischemia-reperfusion injury (MIRI) and trimetazidine (TMZ), supported by relevant and up-to-date references. The context is sufficient to justify the study.

Structure, Figures, and Tables: The article follows a logical structure (Introduction, Methods, Results, Discussion, Conclusion). Figures and tables are well-labeled, high-quality, and support the findings. Supplementary materials (e.g., Table S1-S9, Supplementary Figures) enhance transparency.

Raw Data: The manuscript mentions registration on PROSPERO (CRD42022377728) and includes supplementary files, suggesting raw data availability. However, explicit confirmation of full raw data accessibility (e.g., repository links) would strengthen this aspect.

Self-Contained Results: Results align with the hypotheses, particularly TMZ’s efficacy in reducing oxidative stress markers (SOD, MDA) and infarct size. Subgroup analyses add depth.

Experimental design

Originality and Scope: The study is a systematic review/meta-analysis of preclinical data, fitting within the journal’s scope. It addresses a gap in consolidating evidence on TMZ’s effects in rat MIRI models.

Research Question: Well-defined and meaningful, focusing on TMZ’s therapeutic efficacy. The rationale for using rat models is justified (similar cardiac physiology, reproducibility).

Rigor and Ethics: Methods adhere to PRISMA guidelines, and SYRCLE/CAMARADES tools assess bias/quality. Ethical compliance is noted, though some studies lacked blinding/allocation details (common in animal studies).

Methodological Detail: Search strategy, inclusion/exclusion criteria, and statistical methods (random-effects models, sensitivity analyses) are thoroughly described. Replication is feasible, though heterogeneity sources (e.g., dosing variability) are acknowledged.

Validity of the findings

Impact and Novelty: The study synthesizes disparate preclinical data, offering preclinical evidence for TMZ’s use in MIRI. Novelty lies in subgroup analyses (e.g., optimal dosing: 3–10 mg/kg; ischemia duration <30 min).

Data Robustness: Results are statistically sound, with sensitivity analyses confirming stability. High heterogeneity (e.g., I² = 99% for infarct size) is addressed via subgrouping/meta-regression, though residual heterogeneity suggests caution in interpretation.

Conclusions: Well-supported by results, linking back to the research question. Limitations (e.g., publication bias, animal-to-human translation) are appropriately noted.